# Is Adiposity Associated with the Quality of Movement Patterns in the Mid-Adolescent Period?

**DOI:** 10.3390/ijerph17249230

**Published:** 2020-12-10

**Authors:** Josip Karuc, Goran Marković, Marjeta Mišigoj-Duraković, Michael J. Duncan, Maroje Sorić

**Affiliations:** 1Faculty of Kinesiology, University of Zagreb, 10000 Zagreb, Croatia; goran.markovic@kif.unizg.hr (G.M.); marjeta.misigoj-durakovic@kif.unizg.hr (M.M.-D.); maroje.soric@kif.unizg.hr (M.S.); 2Faculty Research Centre for Sport, Exercise and Life Sciences, Coventry University, Coventry CV1 5FB, UK; aa8396@coventry.ac.uk

**Keywords:** obesity, paediatric exercise, motor control, motor coordination, motor competence, movement competence

## Abstract

This study examined the association between functional movement (FM) and adiposity in adolescent population (16–17 years). This study was conducted in a representative sample of urban adolescents as the part of the CRO-PALS longitudinal study (n = 652). Body mass index (BMI), a sum of four skinfolds (S4S), waist and hip circumference were measured, and FM was assessed via Functional Movement Screen^TM^ (FMS^TM^). Furthermore, total FMS^TM^ screen was indicator of FM with the composite score ranged from 7 to 21, with higher score indicating better FM. Multilevel analysis was employed to determine the relationship between different predictors and total FMS score. In boys, after controlling for age, moderate-to-vigorous physical activity, and socioeconomic status, total FMS score was inversely associated only with BMI (β = −0.18, *p* < 0.0001), S4S (β = −0.04, *p* < 0.0001), waist circumference (β = −0.08, *p* < 0.0001), and hip circumference (β = −0.09, *p* < 0.0001). However, among girls, in adjusted models, total FMS score was inversely associated with S4S (β = −0.03, *p* < 0.0001), while BMI (β = −0.05, *p* = 0.23), waist circumference (β = −0.04, *p* = 0.06), and hip circumference: (β = −0.01, *p* = 0.70) failed to reach statistical significance. Results showed that the association between adiposity and FM in adolescence is sex-specific, suggesting that boys with overweight and obesity could be more prone to develop dysfunctional movement patterns. Therefore, exercise interventions directed toward correcting dysfunctional movement patterns should be sex-specific, targeting more boys with overweight and obesity rather than adolescent girls with excess weight.

## 1. Introduction

Childhood obesity presents one of the largest public health problems with serious long-term health consequences. Children with obesity have a higher risk of developing diabetes type 2, cardiovascular diseases, cancer, and musculoskeletal disorders later in life [1,2,3]. Amongst the above-mentioned risks of paediatric obesity, musculoskeletal disorders with associated biomechanical and health impact have been least studied systematically. According to the most recent umbrella review which investigated the association between adiposity and physical function in children with obesity, few published systematic reviews drew attention to the phenomena of the biomechanics of childhood obesity [4]. Indeed, only one review pointed to the negative influence of excess weight on movement biomechanics among children with possible long-term consequences on musculoskeletal health [5]. This is surprising since musculoskeletal disorders are currently among the most common diseases and one of the leading world health-care problems [6]. For example, direct costs of the musculoskeletal diseases among children in the United States amount to more than 7.6 billion USD per year [7]. In addition to this, it is well known that pathobiomechanical behaviour is a key factor for the development of postural and musculoskeletal pathologies in children and adults [8,9,10,11]. However, it remains unknown what role adolescent obesity plays in the development of different pathobiomechanical movement behaviour.

According to the recent evidence, three inter-related clinical and biomechanical components are compromised in children with overweight: (1) body posture [12,13,14,15,16], (2) gait biomechanics [5,17], and (3) movement competence [4,18,19,20]. Compared to normal weight peers, children with overweight demonstrate postural malalignments characterized by thoracic hyperkyphosis, lumbar hyperlordosis, genu valgum and varus, valgus heel, and flat feet [12,13,14,15,16]. Furthermore, children with overweight have altered biomechanics of gait associated with higher hip and tibiofemoral contact force, and lower limb valgus position [5]. Additionally, during walking, these children exhibit increased maximum force beneath the lateral and medial forefoot, greater pressure-time integral beneath the midfoot and 2nd–5th metatarsal area [17]. Along with the postural abnormalities and altered biomechanics of walking, children with overweight have compromised movement competency [18,19,20]. Movement competency is defined as the global movement patterns (i.e., locomotion, object control skills, or stability tasks) essential for the child motor development [21,22]. An important aspect of the movement competency, and thus of motor development, is the qualitative component of the movement, often termed as the movement quality.

Functional movement (FM) represents a clinical measure of movement quality [23,24,25], most commonly assessed via Functional Movement Screen^TM^ (FMS^TM^) [26,27]. FM implies an optimal range of motion, balance, and postural control of the specific movement [26,27,28]. Contrary, dysfunctional movement (DFM) presents suboptimal movement quality and is related to a compensatory movement pattern along the kinetic chain with associated loss in the range of motion, balance, and deficit in postural control of the specific movement pattern [26,27,28]. The importance of FM patterns has been discussed in previous studies [23,24,25,26,27] and they are considered as fundamental ‘pillars’ for the exhibition of complex movements [26,27], whereas DFM has been related to higher injury incidence [29,30,31,32] and potential movement pathologies in children with overweight [19,20,33]. Therefore, incorporating FM patterns in exercise programs is critical for the optimal progress toward more complex movement skills [26,27,28]. In addition to the aforementioned consequences of obesity on biomechanics, evidence shows that higher body weight changes motor performance, range of motion [34], balance [34] and leads to poor postural control in children [35], which could potentially endanger the performance of both FM and complex movement skills. Evidence suggest that obesity has a high impact on joint structures responsible for joint stabilization and proprioception [36]. Furthermore, obesity leads to degenerative deformities such as osteoarthritis even among children [37]. Taking everything into account, postural alteration, coordination deficits, and overweight in childhood could lead to the development of DFM and orthopaedic deformities in the future [1,19,20]. This could be concerning since the number of children with overweight and obesity is still increasing around the globe [38]. Still, among all previously considered biomechanical components, FM and its relation to the adiposity has not been studied widely.

The development of DFM patterns along with higher body weight through the adolescent period could result in much higher dysfunction on the musculoskeletal system. Moreover, neuromuscular control and movement coordination are not completely developed by the time of adolescence [39]. Therefore, investigating relations between adiposity and FM is important for the musculoskeletal health of the mid-adolescents (14–17 yo) as well. Few studies investigated quality of movement and obesity among the paediatric participants with the majority reporting an inverse relationship between FM and weight status [19,20,34,40,41], while only one study revealed no correlation between these two variables [42]. However, these studies had a small sample size, or recruited merely participants with overweight/obesity, and did not include mid-adolescents. This makes it difficult to ascertain how FM and adiposity might be related during this period, and without this, scientists, physical educationalists, and clinicians may make erroneous decisions by applying outcomes found on children onto mid-adolescents. To the best of our knowledge, there are no studies that examined the association between FM and adiposity in a large, random sample of mid-adolescents. Therefore, this study aims to bridge this gap by examining the sex differences in FM between mid-adolescents with normal weight and mid-adolescents with overweight and obesity. In addition, this study will thrive to reveal the relationship between variables of adiposity and FM in a representative sample of urban mid-adolescents.

## 2. Materials and Methods

### 2.1. Participants

Current study was performed as a part of the 4-year longitudinal study (CRO-PALS) conducted in representative sample of urban youth (city of Zagreb, Croatia). Information about the procedures of the CRO-PALS longitudinal study have been documented in previous research [43]. Shortly, using stratified two-stage random sampling procedures (school level and class level), 54 classes in 14 secondary schools were selected to participate in the CRO-PALS study. All 1408 schoolchildren in the selected classes were approached, and 903 agreed to participate (response rate = 64%). 157 participants were not available on the day of assessment or did not perform the FMS^TM^ procedure and anthropometric measurement. As a result, information of 746 mid-adolescents were collected.

To be included in the analyses, all subjects had to meet specific criteria, namely: (1) not reporting any pain while performing FMS^TM^ assessment, and (2) not being classified as a child with underweight according to International Obesity Task Force criteria [44]. Accordingly, 94 subjects were excluded. At the end, data on 652 participants were analyzed (girls, n = 321, mean age ± SD = 16.6 ± 0.4 yo; boys, n = 331, mean age ± SD = 16.7 ± 0.4 yo). Measurements were taken in 2015, during March, April, and May. Figure 1 represents a flowchart of the participants that were included in the study.

To examine if the representativeness of the CRO-PALS sample has been preserved in the subsample selected for FMS assessment, we compared 652 participants from the current study to the rest of the CRO-PALS participants. These analyses indicated comparable values of moderate-to-vigorous physical activity and sum of four skinfolds (*p* = 0.8 and *p* = 0.6, respectively), while values of BMI, waist circumference, and hip circumference were slightly different between the participants from the current study compared with the rest of participants (BMI: 22.3 vs. 20.7 respectively, *p* < 0.001; waist circumference: 72.8 vs. 71.0, respectively, *p* = 0.008; hip circumference: 97.8 vs. 95.9, respectively, *p* = 0.002).

Having fully informed the children and their parents about the aims of the study, its protocol, and the possible hazards and discomforts related to the procedures used, written consent was obtained from both children and their parents or legal guardians. The study was performed according to the Declaration of Helsinki and the procedures were approved by the Ethics Committee of the Faculty of Kinesiology, University of Zagreb (Croatia) (No: 1009-2014). 

### 2.2. Procedures

#### 2.2.1. Outcomes: Functional Movement Screen Variables

FM (i.e., movement quality) was assessed via FMS^TM^. FMS^TM^ is a screening instrument designed for evaluation of mobility and stability of the seven functional movement patterns through seven tests [26,27]: the deep squat, hurdle step, inline lunge, shoulder mobility, active straight leg raise (ASLR), trunk stability push-up, and rotary stability. According to previous research, two-hour education on using FMS^TM^ is efficient to gain an optimal interrater and intrarater reliability [45]. However, the FMS^TM^ was performed by ten novice trained raters who participated in two-day training performed by an FMS^TM^ certified practitioner. Additionally, two sessions were organized in order to gain the precision and consistency of rater’s testing procedures. Each participant had a maximum of three trials for each FMS^TM^ movement pattern. After that, the highest FMS^TM^ score from the three trials was documented [26,27]. Each test was scored on a three-point scale (1–3), with higher scores indicating better movement quality. Evidence suggest that pain can change control of the movement patterns [46]. Therefore, participants were asked if they felt pain while performing each of the 7 FMS^TM^ movement patterns. Following this, fifty-three participants felt pain during the FMS^TM^ testing procedure, and were not included in the further analyses. In this study, we defined FM as the movement with a given score of 2 or 3 during FMS^TM^ procedure. Furthermore, a score of 1 was given when the participant was unable to perform movement due to the number of movement compensation present which reflects the DFM pattern [26,27,28]. This means that score of 2 and 3 was an indicator of FM, whereas a score of 1 was an indicator of DFM for each of 7 individual FMS tests. In this way, we could calculate the number and proportion of participants that exhibited DFM in each of the 7 individual FMS tests. This was the basic step for analyzing the differences in the proportion of participants that performed DFM between children with normal weight and children with overweight and obesity for each of 7 individual FMS^TM^ tests (i.e., using chi-square tests). Besides, an overall composite score (total FMS score) was calculated with a total FMS score of 21 according to standardized guidelines reported in the literature [26,27]. In this way, the total FMS score was set as a continuous outcome variable for the 2-way ANOVA and multilevel model regression analysis.

#### 2.2.2. Predictors: Body Mass Index, Sum of Four Skinfolds, Waist and Hip Circumference

Participants were weighed barefoot in their shorts and T-shirts with a pre-calibrated portable digital scale to the nearest 0.1 kg. Height was taken to the nearest 0.1 cm using an anthropometer (GPM, Siber-Hegner & Co., Zurich, Switzerland). Then, body mass index (BMI) was calculated as the body weight in kilograms divided by the body height in meters squared (kg/m^2^) [47]. Age and sex-specific BMI cut-off points proposed by the International Obesity Task Force criteria were used to distinguish between children with normal weight and children with overweight and obesity [44]... For the purpose of this study participants were separated into two weight status groups: normal weight and overweight and obese group of children. Skinfold measurements were taken on the right side of the body at the following sites to the nearest 0.2 mm using Harpenden skinfold calliper (British indicators, West Sussex, UK): (1) triceps-at the back of the upper arm, halfway between the acromion process and the olecranon process, (2) biceps-at the front of the upper arm; at the same level as the triceps, (3) subscapular-about 2 cm below the lower angle of the scapula; a diagonal fold, (4) suprailiac-at the iliac crest; in anterior axillary line plane. Sum of all four skinfold measures was taken as the measure of the subcutaneous tissue content. All skinfold measures were taken in triplicate and median values were used for analyses. Waist and hip circumferences were measured manually with non-stretchable tape in a transverse plane at the midpoint between the last rib and the iliac crest, and at the level of the largest lateral extension of the hips, respectively [48]. Skinfold measurements and body circumferences on all participants were performed by a single, skilled lab technician.

#### 2.2.3. Confounders: Moderate-To-Vigorous Physical Activity (MVPA), Socioeconomic Status (SES), and Age

Physical activity level was assessed with the computerized version of the School Health Action, Planning, and Evaluation System (SHAPES) questionnaire [49]. Details about calculation of physical activity variables (i.e., Moderate-to-Vigorous Physical Activity-MVPA) are described in the literature [49]. In the study done by Wong et al. [49], moderate correlation was reported between the MVPA assessed by SHAPES and the results obtained with accelerometer. In addition, results from the SHAPES questionnaire are comparable with other physical activity instruments for mid-adolescents since above-mentioned research showed moderate agreement for MVPA assessed by SHAPES questionnaire [49].

SES was estimated with the following question: “What do you think about your financial situation when you compare yourself to other peers? Think about how much you can afford.”, while the following answers were offered: 1—Much lower than average, 2—Lower than average, 3—Average, 4—Higher than average, 5—Much higher than average. Furthermore, in all further multievel models chronological age was included as a confounder variable (reported in years). In all models, age was centered around the value 17 and coded as “Age-17”.

### 2.3. Data Analysis

First, descriptive analysis for adiposity and confounder variables was conducted for girls and boys separately. Second, to determine the differences between the group of children with normal weight and overweight and obesity in total FMS score, a two-way ANOVA was employed, using weight status and sex as fixed factors. Third, to examine the differences between the group of children with normal weight and overweight in the proportion of individuals who performed DFM, a chi-square test was employed for boys and girls separately. Lastly, multilevel modelling was used to examine the relationship between the variables of adiposity and total FMS score according to the literature [50].

In the current study, authors have applied multilevel modelling methodology in three distinctive steps. First step included building the model at level one (first model). After that, another model was built (second model) which represented end of the first step. Following this, likelihood ratio test (LR test) was used for comparison of these two models (end of second step). In the third step, the model with a better fit was chosen (end of the third step). After the aforementioned three-step process was conducted, a further model was built, and this process was repeated (from first to the third step). In order to get the final model, this procedure was iterative, which yielded in the number of various multilevel regression models. Lastly, the model with the best fit was chosen and presented as the final model [50] (Figure 2). According to the evidence, sex can affect weight status and total FMS score in the adolescent period [51]. For this reason, all analyses were employed for each sex separately. In order to investigate association between different indicators of adiposity and FM, total FMS score was set as the response variable with the included BMI, the sum of four skinfolds, waist circumference, and hip circumference as predictor variables. Age, physical activity level, and SES were included in all models as the confounders since they have been previously shown to influence the total FMS score [20,52].

Results of descriptive statistics are shown as mean values ± SD. To perform multilevel analyses, MLwiN software (v. 3.04) (Centre for Multilevel Modelling, University of Bristol: Bristol, UK, 2019) was used [53]. Furthermore, Statistica software (v. 13.5) (TIBICO Software Inc., Palo Alto, CA, USA) was employed to perform two-way ANOVA, while significance level was set at *p* < 0.05.

## 3. Results

### 3.1. Descriptive Statistics

Within the group of girls with overweight and obesity, eight girls were classified as girls with obesity (i.e., 15%), while within the group of boys with overweight and obesity, 17 boys (20%) were classified as boys with obesity. Table 1 presents the characteristics of the participants stratified by gender.

### 3.2. Differences between Children with Normal Weight and Children with Overweight and Obesity in Total FMS Score

Results of 2-way ANOVA showed significant effect of sex (df = 1, F = 14.14, *p* = 0.00019), weight status (df = 1, F = 14.43, *p* = 0.00016), and sex * weight status interaction (df = 1, F = 8.81, *p* = 0.00016) on total FMS score. After that, Bonferroni’s post-hoc test was performed and revealed a significant difference in total FMS score within the group of boys, where boys with normal weight surpass boys with overweight and obesity in total FMS score (12.6 and 11.1, respectively; *p* < 0.0001). Within the group of children with overweight and obesity, girls with overweight and obesity exhibited better results in total FMS score when compared with boys with overweight and obesity (12.6 vs. 11.1, respectively; *p* < 0.0001). Besides, the analysis showed significant interaction, where girls with normal weight outperformed boys with overweight and obesity (total FMS score: 12.8 vs. 11.1, respectively; *p* < 0.001). Results of 2-way ANOVA representing differences within sex and weight status group as well as the interaction between groups in total FMS score are presented in Table 2.

### 3.3. Differences between Children with Normal Weight and Children with Overweight and Obesity in the Proportion of Individuals That Performed DFM in Each FMS Test

Differences in the proportion (%) of girls with normal weight and girls with overweight and obesity that performed DFM in each FMS test are shown in Figure 3. Interestingly, girls with normal weight showed a lower proportion of DFM compared to girls with overweight and obesity in only one FMS test—shoulder mobility (21% vs. 39%, respectively; *p* = 0.04).

Figure 4 demonstrates differences in the proportion (%) of boys with normal weight and boys with overweight and obesity that performed DFM in each FMS test. Boys with normal weight exhibited better quality of movement and lower proportion of DFM compared to boys with overweight and obesity in most of the FMS tests: deep squat (30% vs. 51%, respectively; *p* < 0.0001), inline lunge (29% vs. 43%, respectively; *p* < 0.0001), shoulder mobility (41% vs. 59%, respectively; *p* = 0.01), trunk stability push-up (33% vs. 62%, respectively; *p* < 0.0001), and rotary stability (32% vs. 59%, respectively; *p* < 0.0001).

### 3.4. Relationship between the Variables of Adiposity and Total FMS Score

Since this study investigated high school children, in the current research, students were at level-1, clustered within classes at level-2 nested within schools at level-3, indicating that data have level-three hierarchical structure. In order to know how much clustering there is in our data, the variation partition coefficient (VPC) was calculated. Significant clustering among both sexes was seen (girls: VPC = 0.919; boys: VPC = 0.865), indicating that 8.1% and 13.5% of total FMS score variation lies within classes, and 91.9% and 86.5% of variation lies between girls and boys, respectively. When VPC is represented as an intraclass correlation coefficient (ICC), the correlation in total FMS score within classes among girls is 0.09, and 0.13 among boys; suggesting that there is non-independence in our dataset (see Appendix A, Figure A1). Level-3 deviance (D3) dropped when the level-3 model was introduced (D3 dropped by 0.25 in girls, and 0.6 in boys); and the level-3 model was therefore not significant (among girls: LR = D2 − D3 = 1460.567 − 1460.317 = 0.25; among boys: LR = D2 − D3 = 1520.92 − 1520.32 = 0.6). However, deviance at level-2 (D2) dropped significantly among both sexes (among girls: LR = D1 − D2 = 1460.145 − 1460.567 = 5.78; among boys: LR = D1 − D2 = 1534.85 − 1520.92 = 13.93). The quantile-quantile plots demonstrated an approximately straight line, implicating that normality assumption was reasonable (both at level-2 and level-1) among girls and boys.

After the initial investigation of hierarchical data, the three-step multilevel modeling approach was carried on as described in Section 2.3 Data Analysis (see also Figure 2). The random-intercept model was built first and included one of the predictors (e.g., BMI) with confounders: age, MVPA, and SES. Next, for each of the predictor variables (e.g., BMI), the random-slope model was introduced. Following this, LR test was used to compare aforementioned models and to choose the model with a better fit [50]. At the end, predictor coefficients were modeled as random at level-2. Accordingly, separate analysis was done for each of the predictors which yielded in eight separate analyses (see Table 3). The coefficients shown in the tables are mean unstandardized coefficients (β) representing association between total FMS score and the different predictors. Although the addition of the confounders did not improve the models, nonsignificant confounders were retained in the models since they can influence the predictor coefficient [54].

#### 3.4.1. Association between Indicators of Adiposity and Total FMS Score among Girls

First, as a part of the multilevel approach, correlation analysis between BMI, the sum of four skinfolds, waist circumference, hip circumference, and total FMS score were employed separately. Results showed significant relationship between total FMS score and sum of four skinfolds (r = −0.199, *p* < 0.0001), while relationship with other predictors failed to reach significance (BMI: r = −0.05, *p* = 0.353; waist circumference: r = −0.095, *p* = 0.091; hip circumference: r = −0.0075, *p* = 0.894) (see Appendix A, Figure A2). After that, multilevel approach was employed. In the evaluation of the association between different adiposity predictors and total FMS score, level-2 random-intercept models were chosen as the models with the best fit in girls. When the models were controlled for chronological age, MVPA, and SES, only the sum of four skinfolds showed a significant association with the total FMS score (β = −0.03, *p* < 0.0001), while the coefficient for waist circumference approached significance (β = −0.04, *p* = 0.06). On the other hand, coefficients for BMI and hip circumference failed to reach significance (β = −0.05, *p* = 0.23; β = −0.01, *p* = 0.70, respectively) (Table 3).

#### 3.4.2. Association between Indicators of Adiposity and Total FMS Score among Boys

Within the group of adolescent boys, correlation analysis showed significant relationship between total FMS score and all predictors (BMI: r = −0.2596, *p* < 0001; sum of four skinfolds: r = −0.327, *p* < 0.0001; waist circumference: r = −0.2639, *p* = 0.0001; hip circumference: r = −0.2775, *p* = 0.0001) (see Appendix A, Figure A3). Following this, the evaluation of the association between BMI, the sum of four skinfolds, waist and hip circumferences with total FMS score, yielded in level-2 random-intercept models as the models with the best fit. When the models were controlled for age, MVPA, and SES, all predictors demonstrated significant associations with total FMS score (BMI: β = −0.18, *p* < 0.0001; sum of four skinfolds: β = −0.04, *p* < 0.0001; waist circumference: β = −0.08, *p* < 0.0001; hip circumference: β = −0.09, *p* < 0.0001) (see Table 3). For instance, in the aforementioned multilevel analysis of the association between total FMS score and BMI in boys, BMI coefficient was −0.18 and it is shared by all school classes. The above-mentioned results can be interpreted as following: “increment of 5-point in BMI corresponds to a near to 1-point decrease in total FMS score among boys, holding other confounding variables constant”.

## 4. Discussion

To our knowledge, this is the first study that has analyzed sex differences in FM between children with normal weight and children with overweight and obesity in the mid-adolescent period. The strength of our approach is seen in large sample size, separate analyses of girls and boys, control of known confounding variables (i.e., age, physical activity level, and SES), and the use of several variables of adiposity in analyses. The key findings of the present study are related to: (a) sex-specific differences in total and individual FMS scores between mid-adolescents with normal weight and mid-adolescents with overweight and obesity, and (b) generally low, but sex-specific, relationship between variables of adiposity and total FMS score in mid-adolescents.

Performance of our participants in FMS was comparable to 13–18-year old high-school athletes [55], but lower than those of Indian adolescents recreationally or competitively participating in sports [51]; however, the studied sample in the latter study had a much wider age range (i.e., 10–17 years). Given that age and maturity status could have a significant effect on the FMS score [56], it could be a plausible reason for the observed discrepancy in findings. In the present study, we did not observe practically relevant sex differences in total FMS score, considering the total sample. This is in contrast to findings obtained on recreational or competitive adolescent athletes [51,57], where boys significantly outperformed girls in total FMS score (mean difference: 0.8–1.5). Thus, it seems that, at least in adolescents, sports participation could significantly affect sex-related differences in total FMS score.

In the current study, boys with overweight and obesity, but not girls, had significantly lower total FMS score compared to their peers with normal weight, suggesting that the association between adiposity and movement quality in mid-adolescents could be sex-specific. This finding is reinforced by the results of multilevel analyses, which showed a statistically significant relationship between variables of adiposity and total FMS score in boys, but not in girls. A previous study conducted by Duncan et al. [19] reported significantly lower FMS scores in overweight/obese group vs. normal weight group of school children (7–10 yo), but did not include a separate analysis for each sex. To date, only one study examined sex dimorphism in regards to adiposity and FM and did not reveal a sex difference in total FMS score within group of children with normal weight, overweight or obesity [34]. Furthermore, the same study did not report a significant difference in total FMS score between group of boys with normal weight and overweight, which is contrary to results reported in the current study. It should be noted that a higher proportion of children with obesity was higher among boys compared to girls in the current study which could partly drive sex differences noted here. However, these findings are difficult to compare since the previously mentioned study included much younger participants (9.6 ± 1.5 yo) with a larger age span (6–13 yo) [34]. Overall, our results suggest, for the first time, that sex is significantly associated with the movement proficiency and variables of adiposity in children and mid-adolescents.

There are two possible explanations behind the observed sex-specific relationship between adiposity and movement quality in mid-adolescents. (1) Neuromechanical: Evidence shows that adolescent boys are more prone to develop postural misalignment, such as hyperkyphosis compared to girls [16,58]; since kyphotic posture decreases concentric activity of the thoracic paraspinal muscles, this could directly limit shoulder mobility test whereas this movement pattern requires active thoracic extension (i.e., demands concentric action of the paraspinal thoracic extensors muscles) [26,27,28]. Moreover, hyperkyphosis can directly limit the optimal performance of the squat, inline lunge, and rotary stability since these patterns demand maintaining neutral spine position and co-contraction of the paraspinal thoracic muscles [26,27,28]. Furthermore, flat feet are mostly seen in boys with overweight rather than in girls with overweight [59], which can cause deficits in uni- and contra- lateral lower-extremity stability movement patterns (i.e., in-line lunge and hurdle step) [26,27,28]. Furthermore, adolescent boys with obesity demonstrate impairment of the rectus femoris muscle activation which is recruited while performing lower extremities movement patterns [60]. (2) Physiological: Sex-specific association between adiposity and movement quality is likely to be the result of the maturation process, which differs between girls and boys and may have different impacts on movement proficiency during childhood and adolescence [61]. Looking altogether, deficits which arose from each FMS^TM^ test resulted in decrease of the total FMS score among boys, but not girls. In the present study, regardless of the measure of adiposity used, common variance between adiposity and total FMS score did not exceed 10%. Previous studies have reported considerably larger inverse associations between BMI and FMS scores then in the current study (*r* ranged from −0.3 to −0.81) [19,20,34,40,41]. Notably, several studies have reported that the strength of the association between BMI and motor coordination declines as children start to reach puberty [62,63], which might explain some of the above-mentioned disparity in findings between the current and previous research.

Although our results shed some new light on the association of adiposity and movement quality of mid-adolescents, measured with the total FMS score, this approach could be methodologically limited, primary because of the poor factorial validity of the total FMS score. Several exploratory factor analyses of individual FMS tests have consistently reported in both youth and adults that the 7 tasks of FMS^TM^ have low internal consistency and were not indicators of a single factor [64,65,66]. Indeed, a 2-factor structure of individual FMS scores has been always observed, with extracted factors explaining only 38–47% of the variance of all 7 FMS^TM^ tests, and with inconsistent and non-interpretable factor structure among different populations. As a result, the sum score of all FMS^TM^ tests does not represent a consistent and valid measure of human movement quality. This is particularly problematic during growth and maturation when differentiation of general movement coordination and motor qualities are occurring [55].

When the focus is shifted to individual FMS^TM^ tests, the limitation of particular tests, and consequently, the total FMS score to discriminate adolescents of different body size or composition becomes obvious. First, both our and several previous studies [51,55,57] have shown that female participants exhibit the worst performance in trunk stability push up, which is essentially a trunk and upper-body strength test, not a trunk stabilization test. In our case, 80% of girls with normal weight failed to perform trunk stability push up correctly, suggesting that the test was too difficult even for girls with normal body weight (Figure 3). Second, ASLR, which is essentially a flexibility test of the posterior leg, was too easy for adolescent girls, as only about 5% of them had a dysfunctional pattern in that test (Figure 3). As a result, both tests failed to contribute to the discriminatory power of the total FMS score when comparing adolescent girls with different body compositions. Of the remaining five FMS^TM^ tests, only the shoulder mobility test significantly discriminated adolescent girls with normal weight and girls with overweight. Notwithstanding the possibility that movement quality in adolescence is indeed less affected by adiposity in females, results of this large cross-sectional study question the usefulness of some FMS^TM^ tests in assessing movement quality of adolescent girls. In contrast, in boys, 5 out of 7 FMS^TM^ tests discriminated between those with normal weight and those with overweight and obesity (Figure 4).

Still, the general question remains: “How could overweight and obesity contribute to nonoptimal movement quality and what are the potential consequences of DFM in mid-adolescence period?”. The possible answer could give a concept which is modified according to the model proposed by Page et al. [67]. This modified concept (i.e., model of musculoskeletal dysfunction cycle and contribution of obesity in the potential development of DFM patterns) consists of the seven stages (Figure 5). Evidence shows that higher body weight puts additional load on the joints [36], which, along with the sedentary behaviour and physical inactivity, can create muscle imbalances in overweight children [60] (stage 1) and lead to poor postural control [35] (stage 2). Compromised posture is associated with DFM patterns in overweight children [33] (stage 3). These DFM obesity-associated patterns could result in faulty motor program (stage 4) and altered proprioception (stage 5) with possible long-term consequences on musculoskeletal health, mainly joint degeneration [37] (stage 6) and body pain (stage 7). All aforementioned stages are presented as the separate elements within the presented model, where DFM, obesity, and gender could hypothetically play a significant role for the development of the postural pathologies in mid-adolescents. Accordingly, the current study could add the important piece of information within this cycle. However, further longitudinal and intervention studies are required to investigate impact of obesity on FM and musculoskeletal health in mid-adolescent population. Inclusive, both overweight and obesity negatively affect the quality of FM in mid-adolescents which could impact musculoskeletal health later in life. Therefore, exercise interventions that target both obesity and DFM needs to be incorporated into youth physical activity programs, school’s curriculum, and youth sports.

### Strengths and Limitations

There are many key strengths of current research that should be pointed out. To the best of the authors knowledge, this is the only research that has examined the association between FM (i.e., movement quality) and adiposity in the general population of mid-adolescents. Second, a large, age-homogeneous sample of mid-adolescents were recruited in this research (n = 652). Third, this is the first study that has employed the multilevel modelling for prediction of the FM through different indicators of adiposity in the population of mid-adolescents. This approach led to less biased conclusions. Fourth, this is the only research that has investigated relationship between adiposity variables and FM in mid-adolescent for boys and girls separately. This approach gave a new and valuable information about sex-specific relationships between FM and adiposity variables. Fifth, in this study adiposity was represented via four different indicators which can give deeper insight into relation between adiposity and FM. Last, this study controlled three common confounders which can yield in more precise forecast of FM (age, MVPA, and SES). Furthermore, there are few limitations in this study. First, ten raters performed FMS^TM^ procedure which could affect consistency of the results. However, all ten raters went through uniform training on FMS^TM^ protocol, while recent evidence demonstrated good level of interrater agreement in the FMS^TM^ scores [45,68]. Furthermore, a higher proportion of children with obesity among boys compared to girls, could drive some of the sex differences in the association of adiposity and FM reported here. However, given that this is a population-based study, this sex-difference in obesity rate was expected because they are based on the available data from the recent collation of representative studies in Croatian mid-adolescents [69]. Lastly, participants from an urban area were recruited, while excluding rural mid-adolescents, which can influence generalizability of the findings.

## 5. Conclusions

Up to this point, there are no studies that have examined the association between a variety of adiposity indicators and FM in the population of mid-adolescents. Therefore, results of the present study extent previous knowledge on another population. The current study presents novel data demonstrating that overweight and obesity could possibly have detrimental effect on movement quality, as assessed by the FMS^TM^, in boys, but not girls. Of particular note, although FMS scores were poorer in boys with overweight compared to boys with normal weight, being overweight and obese did not appear to confer the same detriment on quality of movement in girls. This means that the association between adiposity and FM in adolescence is sex-specific, suggesting that boys with overweight and obesity could be more prone to develop dysfunctional movement patterns. Development of the optimal movement quality during adolescence is crucial step towards improving musculoskeletal health among girls and boys with overweight and obesity. More precisely, exercise interventions directed toward correcting dysfunctional patterns should be, to some degree, sex-specific, targeting more boys with overweight and obesity than girls with overweight. However, practicing FM patterns in both girls and boys with overweight and obesity could be helpful in order to potentially reduce the injury risk and future development of the postural pathologies in course of the adulthood. Overall, the results of the current study provide important information on the potential negative impact of adiposity on movement quality in the population of average mid-adolescents. This information is particularly important for physical therapists and coaches in youth sport, physical education teachers, and those working in the public health sector.

## Figures and Tables

**Figure 1 ijerph-17-09230-f001:**
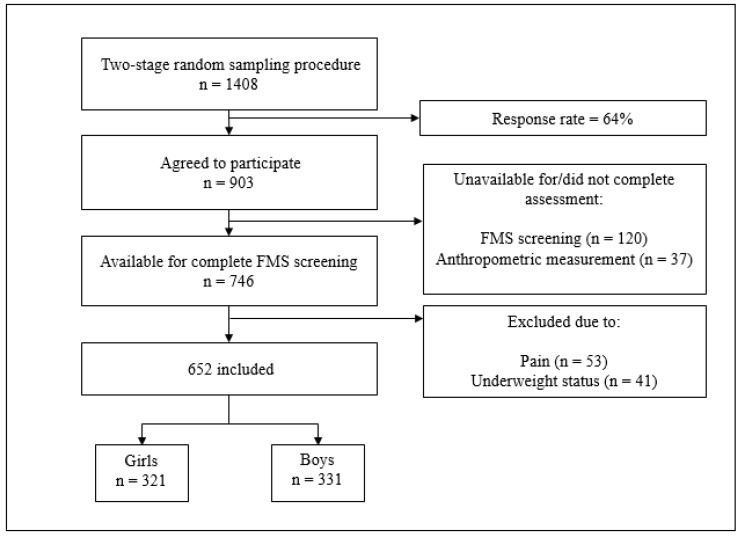
Flowchart of included participants. Note: FMS screening = functional movement screening standardized procedure.

**Figure 2 ijerph-17-09230-f002:**
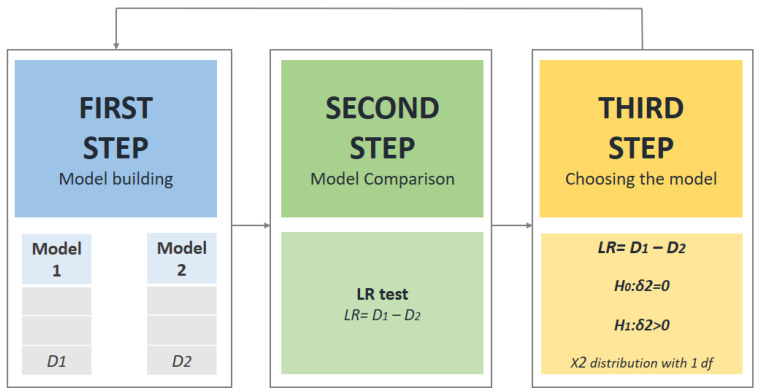
Three-step model building methodology.

**Figure 3 ijerph-17-09230-f003:**
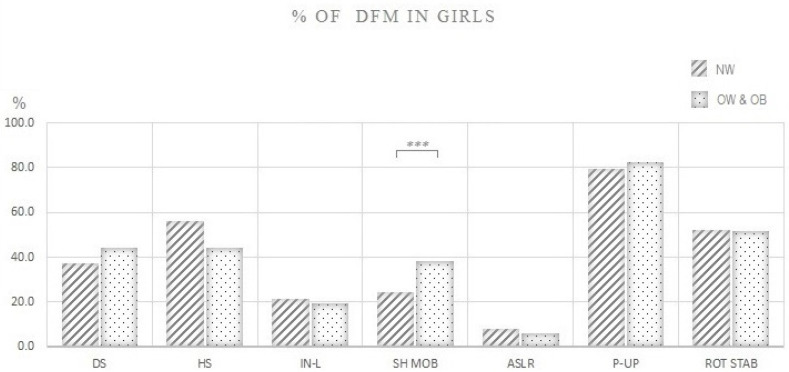
Proportion (%) of girls with normal weight and girls with overweight and obesity that performed dysfunctional movement (DFM) in each FMS test. Note: % OF DFM IN GIRLS: Proportion (%) of girls that performed dysfunctional movement; NW: Girls with normal weight; OW & OB: Girls with overweight and obesity; DS: Deep squat; HS: Hurdle step; IN-L: Inline lunge; ASLR: Active straight leg raise; SHO MOB: Shoulder mobility; P-UP-: Trunk stability push-up; ROT STAB: Rotary stability. *** *p* = 0.04.

**Figure 4 ijerph-17-09230-f004:**
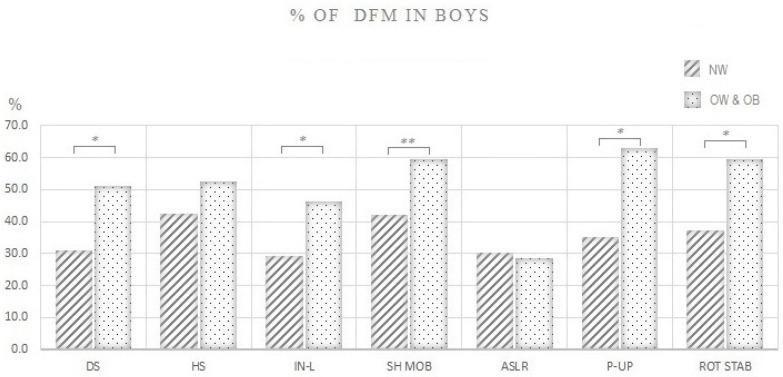
Proportion (%) of boys with normal weight and boys with overweight and obesity that performed dysfunctional movement (DFM) in each FMS test. Note: % OF DFM IN BOYS: Proportion (%) of boys that performed dysfunctional movement; NW: Boys with normal weight; OW & OB: Boys with overweight and obesity; DS: Deep squat; HS: Hurdle step; IN-L: Inline lunge; ASLR: Active straight leg raise; SHO MOB: Shoulder mobility; P-UP: Trunk stability push-up; ROT STAB: Rotary stability. * *p* < 0.0001, ** *p* = 0.01.

**Figure 5 ijerph-17-09230-f005:**
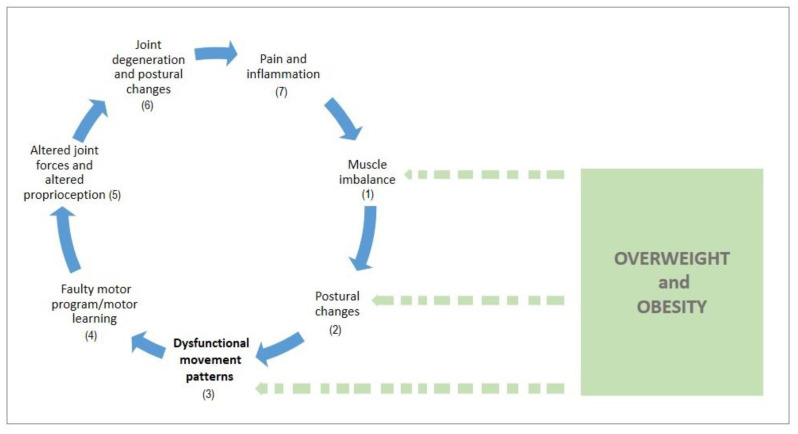
Model of musculoskeletal dysfunction cycle and contribution of obesity in the potential development of DFM patterns (modified according to Page et al., 2010).

**Table 1 ijerph-17-09230-t001:** Characteristics of the study sample stratified by gender.

Descriptive Characteristics	Girls (n = 321)	Boys (n = 331)
Age mean (SD)	16.6 (0.4)	16.7 (0.4)
OW & OB within each group n (%)	52 (16.2)	84 (25.4)
BMI (kg/m^2^) mean (SD)	22.1 (3.0)	22.6 (3.4)
Sum of four skinfolds (mm) mean (SD)	49.9 (14.8)	37.1 (17.8)
Waist circumference (cm) mean (SD)	69.3 (6.2)	76.2 (7.3)
Hip circumference (cm) mean (SD)	97.5 (7.2)	98.1 (7.3)
MVPA (min/day) median (IQR)	85.7 (73.5)	117.9 (86.3)
SES median (IQR)	3 (1)	2 (1)

OW: overweight; OB: obesity; OW & OB within each group: Number (n) and percentage (%) of participants with overweight and obesity classified according to the International Obesity Task Force cut-offs within each sex group; BMI: Body Mass Index; MVPA: Moderate-to-Vigorous Physical Activity; SES: socio-economic status; IQR: Interquartile Range; SD: Standard Deviation.

**Table 2 ijerph-17-09230-t002:** Results of 2-way ANOVA representing differences within the sex and weight status group as well as the interaction between groups in total FMS score.

Sex/Weight Status Group	NW	OW & OB	Within Sex Group Difference	Interaction
Girls	12.8	12.6	*p =* 1.00	*p =* 1.00 ^1^
Boys	12.6	11.1	*p <* 0.0001	*p <* 0.001 ^2^
within weight status group difference	*p =* 1.00	*p <* 0.0001		

OW: overweight; OB: obesity; NW: Group of children with normal weight status; OW & OB: Group of children with overweight and obesity; Interaction: ^1^
*p*-value for the difference between group of boys with NW and group of girls with OW & OB; ^2^
*p*-value for the difference between group of girls with NW and boys with OW &OB; Post-hoc results were obtained using the Bonferroni post-hoc test.

**Table 3 ijerph-17-09230-t003:** Relationship between variables of adiposity and total FMS score among girls and boys.

Response	Total FMS Score											
Predictor	BMI			Sum of Four Skinfolds			Waist Circumference			Hip Circumference		
Parameter	β	S.E.	*p*	β	S.E.	*p*	β	S.E.	*p*	β	S.E.	*p*
Girls	−0.05	0.04	0.23	**−0.03**	**0.01**	**0.0001**	−0.04	0.02	0.06	−0.01	0.02	0.70
Boys	**−0.18**	**0.04**	**0.0001**	**−0.04**	**0.01**	**0.0001**	**−0.08**	**0.02**	**0.0001**	**−0.09**	**0.02**	**0.0001**

Total FMS score: total Functional Movement Screen score; BMI: Body Mass Index; β: beta unstandardized coefficient; S.E.: Standard Error; *p*: *p*-value. All models are adjusted for age (centered around the value of 17), MVPA, and SES; The bold are significant results.

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
