# Peer review of "Is Adiposity Associated with the Quality of Movement Patterns in the Mid-Adolescent Period?"

_ijerph, 2020, doi:10.3390/ijerph17249230_

Round 1
Reviewer 1 Report
General comments: The authors investigated the association between the obesity status, sex and functional movement screen (FMS) score in mid-adolescent boys and girls. They concluded that obesity status is associated with FMS score sex-dependently. This is a valuable study that may add more evidence for the significance of weight management in order to prevent the future functional health issues (i.e., motor skills). In general, the data are presented and discussed well. I wonder if there is data on physical activity. Since physical activity may directly correlate with FMS score, this may explain some of the discrepancies in analyses between girls and boys. Moreover, it is not clear what criteria was used to define “lean” and “OW&OB” groups. My understanding is that the BMI was used; however, it is well established that BMI is not the most reliable measure of adiposity. Rather the skin folding or waist circumference are the better determinants of adiposity. I would suggest editing the manuscript more thoroughly, there are many repetitions of the data analyses approaches throughout the manuscript (i.e., in the method and result sections). The discussion can be edited to make it more concise. Minor language editing would be suggested. In statistics, the proper terms are “direct” and “inverse” associations/correlations not “positive” and “negative”, respectively.
Detailed comments:
Abstract:
- Lane 18: Define “FMS score”.
- Lane 18-20: This sentence is difficult to understand, I would suggest editing it to something like the following, if I understand the meaning correctly: “In boys, after controlling for age, moderate-to-vigorous physical activity, socioeconomic status and relationship between variables of adiposity, the total FMS score was significantly and inversely correlated with BMI (b=-….), S4S (….), waist circumference (…), and hip circumference (…).
- Lanes 21-24: I would suggest similar edits for the data on girls.
- Lane 24: “Results showed that the effect…” – the presented data do not support that there is an “effect” of adiposity on FM, rather an association between the parameters. Thus, I would suggest using any synonyms of “association” rather than “effect”.
- Lane 24-26: This conclusion does not reflect the title of the manuscript, and also see the comment #5. In general, this comment is valid the conclusions throughout the manuscript.
- Lanes 26-28: This conclusion maybe premature because, as mentioned in the general comments, there is no information on physical activity as well as BMI, if it was used to determine NW and OW+OB groups, is not the best determinant of adiposity.
Introduction:
- Lane 41: Use “weight” or “obesity” instead of “overweight”, if this sentence refers not only to overweight but obese status as well. In general, in many sentences the word “overweight” is being used. Does this mean that the referred data are about overweight children only (lanes 52, 55, 72, etc), and that the authors studied only overweight children? See comment #3 in Materials and Methods, and #1 in the Results comment sections.
- Lane 42-44: What does this sentence refer to, surprising about 1 review article only or the scientific information?
- The term “mid-adolescents” is being used in the introduction. I would suggest adding the age range in parenthesis after the first use of this term, and use this term not “adolescents” throughout the manuscript (e.g., lane 99).
Materials and Methods:
- Lane 141: “a three-point scale (0-3)”, the description in the below lanes 145-147 suggests that only values of 1, 2 and 3 were used, if this is the case this scale should be “1-3”.
- Lanes 166-168: The sentence that starts “Within the group of girls…” belongs the result section.
- Lanes 162-166: Was BMI used as a criterion to define the groups? If so, please specify the cut-offs, and perhaps define the groups (e.g., “NW”, “OW+OB” (Tables), and use those definitions throughout the manuscript.
- How old are these children? Add the data.
Results:
- Table 1: OW & OB – define (see the notes for Table 2). The definitions “NW” and “OW&OB” can be introduced in the introduction and used throughout the manuscript. The criteria and cut-offs can be introduced in the methods section (see comment #3 in the Materials and Methods comment section.
- Lanes 260-262: This sentence is not required since the previous sentence says “in only one FMS test”. Or “However” needs to be removed.
- What is the average FMS for boys and girls?
Minor edits:
- Lane 44: “one of the leading world health-care problems”.
- “boys/girls with overweight” is not a common terminology.
- Lanes 486: replace “obesity” with “obese”.
Author Response
General comment to the Editor and Reviewers:
First of all, thank you for considering our manuscript for revisions to your journal International Journal of Environmental Research and Public Health. All suggestions from the Editor and Reviewers are accepted and incorporated into the manuscript. Also, content changes can be seen within the manuscript within the function ‘track changes’ while some grammatical changes are not explicitly highlighted due to a number of errors through text which could impair the flow and readability of the text. Below are specific answers to the Editor and Reviewers comments.
Reviewer #1
Comments and Suggestions for Authors
General comments: The authors investigated the association between the obesity status, sex, and functional movement screen (FMS) score in mid-adolescent boys and girls. They concluded that obesity status is associated with FMS score sex-dependently. This is a valuable study that may add more evidence for the significance of weight management in order to prevent future functional health issues (i.e., motor skills). In general, the data are presented and discussed well. I wonder if there is data on physical activity. Since physical activity may directly correlate with FMS score, this may explain some of the discrepancies in analyses between girls and boys. Moreover, it is not clear what criteria were used to define “lean” and “OW&OB” groups. My understanding is that BMI was used; however, it is well established that BMI is not the most reliable measure of adiposity. Rather the skin folding or waist circumference are the better determinants of adiposity. I would suggest editing the manuscript more thoroughly, there are many repetitions of the data analyses approaches throughout the manuscript (i.e., in the method and result sections). The discussion can be edited to make it more concise. Minor language editing would be suggested. In statistics, the proper terms are “direct” and “inverse” associations/correlations not “positive” and “negative”, respectively.
Response to general comment:
Thank you for your overall comment about our study and for recognizing the novelty of our research. The authors of the manuscript agree with the reviewer that physical activity is an important factor that can contribute to the difference in the total FMS score between adolescent girls and boys. In line with this, authors have already published article that examined this phenomenon in the mid-adolescence period which further emphasize the importance of the association between functional movement and physical activity level (1) (please visit do: 10.1016/j.ptsp.2020.09.006 ). For this reason, we have already included physical activity level as the covariable in the original manuscript (i.e. moderate-to-vigorous physical activity level - MVPA) in all our regression models for the girls and boys subgroup (please see ‘Methods and Materials’ section, pg.6, lines:188-195).
The authors of this manuscript agree with the reviewer that BMI cut-offs are not perfect. However, according to the world obesity federation (https://www.worldobesity.org/about/about-obesity/obesity-classification), age and sex-specific cut-off values are a reliable measure for discriminating aforementioned groups (boys:http://s3-eu-west-1.amazonaws.com/wof-files/New_Cut_off_Points_Male_Children.pdf;girls:http://s3-eu-west-1.amazonaws.com/wof-files/New_cut_off_points_female_children.pdf). We already included four different indicators of adiposity (BMI, sum of four skinfolds, waist circumference, and hip circumference) in all multilevel regression models (please see pg.5; lines: 167-185). In this way, adiposity was represented via four different anthropometric measures (i.e. indicators). This is further discussed in the comments below (please see responses no.1, 2., and 3.) while description and literature regarding cut-off was already cited within the original manuscript (2).
We appreciate the reviewers’ suggestions to improve the discussion section. Also, we agree that this part of the article should be more concise and shorter in words. However, in the current paper, the authors wanted to emphasize the importance of behind mechanisms in detail because prior work on similar topics failed to explain the findings. Additionally, the authors offered a hypothetical model of musculoskeletal dysfunction cycle and contribution of obesity in the potential development of dysfunctional movement patterns (please see pg.15, lines: 444-462). Therefore, the authors of this manuscript consider that this part is crucial for practitioners as well as researchers for future studies in this field.
Thank you for spotting minor language errors throughout the text. This suggestion further improved our manuscript and the readability of the text. We made a grammatical correction along with the proofreading through the whole manuscript. However, not all grammatical changes are not explicitly highlighted due to a number of errors through text which could impair the flow and readability of the paper. Lastly, we accepted the reviewer's suggestion about the proper usage of the statistical term. These changes have been made through the manuscript. Below are specific answers to the Editor and Reviewers comments.
References regarding general comment:
- Karuc, J., Mišigoj-Duraković, M., Marković, G., Hadžić, V., Duncan, M. J., Podnar, H., & Sorić, M. (2020). Movement quality in adolescence depends on the level and type of physical activity. Physical therapy in sport: official journal of the Association of Chartered Physiotherapists in Sports Medicine, 46, 194–203. https://doi.org/10.1016/j.ptsp.2020.09.006
- Cole, T.J.; Bellizzi, M.C.; Flegal, K.M.; Dietz, W.H. Establishing a standard definition for child overweight and obesity worldwide: International survey. Br. Med. J. 2000, 320, 1240–1243. doi: 10.1136/bmj.320.7244.1240.
Detailed comments and responses:
Abstract:
1. Lane 18: Define “FMS score”.
Response 1: Thank you for this suggestion. The correction has been made in the abstract section (please see pg.1.; line:17-18).
2. Lane 18-20: This sentence is difficult to understand, I would suggest editing it to something like the following, if I understand the meaning correctly: “In boys, after controlling for age, moderate-to-vigorous physical activity, socioeconomic status and relationship between variables of adiposity, the total FMS score was significantly and inversely correlated with BMI (b=-….), S4S (….), waist circumference (…), and hip circumference (…).
Response 2: Thank you for spotting this. The correction has been made accordingly (please see pg.1.; lines: 19-22).
3. Lanes 21-24: I would suggest similar edits for the data on girls.
Response 3: Thank you for the comment. The correction has been made accordingly (please see pg.1.; lines:22-25 ).
4. Lane 24: “Results showed that the effect…” – the presented data do not support that there is an “effect” of adiposity on FM, rather an association between the parameters. Thus, I would suggest using any synonyms of “association” rather than “effect”.
Response 4: The suggestion has been accepted and correction made in the revisited manuscript (please see pg.1.; lines:25-27).
5. Lane 24-26: This conclusion does not reflect the title of the manuscript, and also see the comment #5. In general, this comment is valid the conclusions throughout the manuscript.
Response 5: Thank you for this important note. The authors made improvements in this part of the abstract and through the manuscript as well (please see pg.1.; lines:25-27 ). Accordingly, we used the term ‘excess weight’ as a more precise and appropriate term.
6. Lanes 26-28: This conclusion may be premature because, as mentioned in the general comments, there is no information on physical activity as well as BMI, if it was used to determine NW and OW+OB groups, is not the best determinant of adiposity.
Response 6: The authors of the current manuscript appreciate this comment. Also, the authors agree with the reviewers’ opinion that the conclusion is maybe premature. However, the authors of the current paper remind that in the originally submitted article there was information about physical activity status (please see pg.6.; lines: 188-195). Also, physical activity level (i.e. moderate-to-vigorous physical activity - MVPA) was further included within all multilevel regression models as the covariable (page 11; lines: 3112-314).
Regarding the second part of the comment that concerns BMI, the authors of this manuscript agree with the reviewer that BMI cut-offs are not perfect. However, as stated in the general comment (please see pg.5.; lines:171-173), according to the World Obesity Federation sex-specific cut-off values are proposed as the valuable cut-offs for discriminating weight status groups of children age between 2 to 18 (please visit: https://www.worldobesity.org/about/about-obesity/obesity-classification). Indeed, cut-off values are age and sex-depended (Please see specific cut-offs for boys: http://s3-eu-west-1.amazonaws.com/wof-files/New_Cut_off_Points_Male_Children.pdf; and girls: http://s3-eu-west-1.amazonaws.com/wof-files/New_cut_off_points_female_children.pdf). This means that specific cut-off values are given month-by-month for boys and girls separately. Also, we already included four different indicators of adiposity (BMI, the sum of four skinfolds, waist circumference, and hip circumference) in all multilevel regression models (please see pg.5.; lines:167-185; this can also be seen in the ‘Results’ section ). In this way, adiposity was represented via four different anthropometric measures (i.e. indicators), not only with BMI. Therefore, in this manuscript, adiposity was represented via multiple indicators which should reinforce the results and conclusion drawn.
Introduction:
1. Lane 41: Use “weight” or “obesity” instead of “overweight”, if this sentence refers not only to overweight but obese status as well. In general, in many sentences, the word “overweight” is being used. Does this mean that the referred data are about overweight children only (lanes 52, 55, 72, etc), and that the authors studied only overweight children? See comment #3 in Materials and Methods, and #1 in the Results comment sections.
Response 1: Thank you for this comment and suggestion. The authors of this manuscript are aware of this issue. However, the terminology that has been used in the current study is in accordance with each study that was referenced. This means that when we cited the study that investigated only overweight children, the term: ‘children with overweight’ was used. This response is in line with the response for comment #3 in Materials and Methods, and #1 in the Results comment sections.
2. Lane 42-44: What does this sentence refer to, surprising about 1 review article only or the scientific information?
Response 2: Thank you for spotting this! This sentence refers to 1 review article only. A slight change was made within the sentence in order to gain better clarity of the sentence (please see pg.2.; lines:42-43).
3. The term “mid-adolescents” is being used in the introduction. I would suggest adding the age range in parenthesis after the first use of this term, and use this term not “adolescents” throughout the manuscript (e.g., lane 99).
Response 3: Thank you for spotting this issue through the text. The age range has been added in the introduction part (please see pg.2.; line: 90). Also, the term “mid-adolescents” was introduced throughout the rest of the manuscript.
Materials and Methods:
1. Lane 141: “a three-point scale (0-3)”, the description in the below lanes 145-147 suggests that only values of 1, 2, and 3 were used, if this is the case this scale should be “1-3”.
Response 1: Thank you for this note. This comment has been accepted and incorporated within the text (please see pg.5., line:150).
2. Lanes 166-168: The sentence that starts “Within the group of girls…” belongs the result section.
Response 2: Thank you for spotting this mistake. This sentence has been deleted from the ‘Materials and Methods’ section and now is incorporated within the ‘Results’ part of the revisited manuscript (please see pg.7.; lines: 235-237).
3. Lanes 162-166: Was BMI used as a criterion to define the groups? If so, please specify the cut-offs, and perhaps define the groups (e.g., “NW”, “OW+OB” (Tables), and use those definitions throughout the manuscript. –world obesity federation cut-off
Response 3: We appreciate the reviewers’ comment. In the current manuscript, BMI was used as a criterion to define the groups according to the International Obesity Task Force (IOTF criteria) as mentioned in the originally submitted manuscript (please see pg.5.; lines:171-173) and above-mentioned response (please see Response 6 Abstract). However, according to the world obesity federation (please see: https://www.worldobesity.org/about/about-obesity/obesity-classification), age and sex specific BMI cut-off points was used (Please see specific cut-offs for boys:http://s3-eu-west-1.amazonaws.com/woffiles/New_Cut_off_Points_Male_Children.pdf; girls:http://s3-eu-west-1.amazonaws.com/wof-files/New_cut_off_points_female_children.pdf ).
In line with this, in the originally submitted manuscript, cut-off points were determined and described in the 'Materials and Methods' section (please see pg.5.; line:171-173).
4. How old are these children? Add the data.
Response 4: Thank you for this comment. According to the reviewers’ suggestion, information regarding age was incorporated within the table in the revisited text (please see Table 1, pg.8.; line:239). However, information about age was noted in the original manuscript within the ‘Participants’ section (please see pg.3.; line:118-119).
Results:
1. Table 1: OW & OB – define (see the notes for Table 2). The definitions “NW” and “OW&OB” can be introduced in the introduction and used throughout the manuscript. The criteria and cut-offs can be introduced in the methods section (see comment #3 in the Materials and Methods comment section.
Response 1: Thank you for this important comment. The definitions and descriptions are now incorporated within Table 1 (below the table) (Please see Table 1; pg.8.; line:240). The criteria and cut-offs are also mentioned at the bottom of the table (see Table 1.; pg.8.; line:242-243).
Comment about The criteria and cut-offs which considered specific response is provided within Response 3 above and within the General response to the reviewer at the first part of this document as well as within the Response 6 of Abstract part.
2. Lanes 260-262: This sentence is not required since the previous sentence says “in only one FMS test”. Or “However” needs to be removed.
Response 2: Thank you very much for suggesting this change. The sentence: ‘However, there was no difference in the proportion of individuals that performed DFM between girls with normal weight and girls with overweight and obesity in other FMS tests.’ is now removed from this part of the ‘result’ section.
3. What is the average FMS for boys and girls?
Response 3: Thank you for asking this question. The average total FMS score for girls and boys separately is not reported within this manuscript since this paper does not investigate average mid-adolescent girls and boys separately. However, there is information about average total FMS score for girls and boys which are within the normal weight or overweight and obese category (total FMS score for girls with normal weight = 12.8; total FMS score for girls with overweight and obesity = 12.6; total FMS score for boys with normal weight = 12.6; total FMS score for boys with overweight and obesity = 11.1) (please see table 2.; pg.9.).
Minor edits:
1. Lane 44: “one of the leading world health-care problems”.
Response 1: Thank you for spotting this! The correction has been made (please see pg.2.; line:45).
2. “boys/girls with overweight” is not a common terminology.
Response 2: Thank you for this comment. However, according to the current guidelines of the Obesity Action Coalition - OAC (see: Home Page - Obesity Action Coalition), the authors used people-first language(please visit: People-First Language - Obesity Action Coalition).
3. Lanes 486: replace “obesity” with “obese”.
Response 4: Thank you for this comment. This change has been made in the revisited manuscript (please see pg.16.; line:498).
Reviewer 2 Report
The authors completed a cross-sectional study to investigate the sex differences in functional movement between adolescents with normal weight and overweight/obese adolescents 16-17 years old. This is a very interesting study that provides new information in the area. The reporting of the evidence included in the paper is complete, as the interpretation of the outcomes. The authors also take into consideration the limitations of their methodology in the elucidation of their outcomes as well as in their conclusion. The paper also includes additional information, which help the reader to consider the outcomes from different angles. I do have however, some minor comments, which you can see below:
- Even though the sample size seems large, there is no information of a prior sample size calculation. This is important, given that it may have affected the statistical analyses and consequently the outcomes. In this regard, the authors report that the range of percentage of the overweight/obese participants is 16.2-25.4%, while a random selection of the participants was used. An elaboration is needed whether there was a matching during the selection process of participants or a randomization process, so that the number of the overweight/obese participants is proportional to the general population. I acknowledge that the authors already report this as a limitation, but for clarity reasons more information is needed.
- Lines 159-177. The skinfold equation used is not mentioned.
- Lines 236-237. F values should also present the degree of freedom.
- Lines 315-326. Usually, when p values are more than 0.05, they are reported as p>0.05.
- There are several spelling and grammar errors in the paper, which require correction (line numbers: 14, 18-21, 95, 100, 105, 136, 216, 222, 242, 305, 307, 308, 356, 416, 446, 466, 476, 490, figures A1-A2 “summ”.
6. Definitions of some acronyms are required within the text and tables. Particularly in tables, these should present outcomes without to be required checking the text, therefore, the definitions are necessary.
Author Response
General comment to the Editor and Reviewers:
First of all, thank you for considering our manuscript for revisions to your journal International Journal of Environmental Research and Public Health. All suggestions from the Editor and Reviewers are accepted and incorporated into the manuscript. Also, content changes can be seen within the function ‘track changes’ while some grammatical changes are not explicitly highlighted due to the number of errors through text which could impair the flow and readability of the text. Below are specific answers to the Editor and Reviewers comments.
Reviewer #2
Comments and Suggestions for Authors
The authors completed a cross-sectional study to investigate the sex differences in functional movement between adolescents with normal weight and overweight/obese adolescents 16-17 years old. This is a very interesting study that provides new information in the area. The reporting of the evidence included in the paper is complete, as the interpretation of the outcomes. The authors also take into consideration the limitations of their methodology in the elucidation of their outcomes as well as in their conclusion. The paper also includes additional information, which help the reader to consider the outcomes from different angles. I do have however, some minor comments, which you can see below:
Response to general comment: Thank you very much for your general comment about our research!
1. Even though the sample size seems large, there is no information of a prior sample size calculation. This is important, given that it may have affected the statistical analyses and consequently the outcomes. In this regard, the authors report that the range of percentage of the overweight/obese participants is 16.2-25.4%, while a random selection of the participants was used. An elaboration is needed whether there was a matching during the selection process of participants or a randomization process, so that the number of the overweight/obese participants is proportional to the general population. I acknowledge that the authors already report this as a limitation, but for clarity reasons more information is needed.
Response 1: Thank you very much for this important comment. In the current manuscript, it has been stated that this study is conducted in a representative sample of urban youth (please see pg.3.; lines:106-107). Results are based on the random sample of urban youth in Zagreb (Croatia) which has been pointed out in the original manuscript as well (please see pg.3; lines:109-112). Also, additional analyses and results regarding representativeness have been included in the revisited manuscript (pg.4.; lines: 122-129).
However, given that this is a population-based study, this sex-difference in obesity rate was expected because they are based on the available data from the recent collation of representative studies in Croatian mid-adolescents (please see: http://www.thelancet.com/journals/lancet/article/PIIS0140-6736(20)31859-6/fulltext). Indeed, the authors agree with the reviewer that more information is needed. Therefore, an explanation about this limitation (please see pg.15.: line:486-488) as well as the reference that supports the aforementioned notion has been incorporated within text (please see pg.21.; line: 729-732).
Reference:
NCD Risk Factor Collaboration (NCD-RisC). Height and body-mass index trajectories of school-aged children and adolescents from 1985 to 2019 in 200 countries and territories: a pooled analysis of 2181 population-based studies with 65 million participants. Lancet 2020, 396, 1511–1524. doi: 10.1016/S0140-6736(20)31859-6
2. Lines 159-177. The skinfold equation used is not mentioned.
Response 2: Thank you for spotting this mistake! We added a sentence that describes the calculation (please see pg.5.; lines: 180-181).
3. Lines 236-237. F values should also present the degree of freedom.
Response 3: Thank you for this important note. The correction has been made and information about degrees of freedom incorporated into the manuscript (please see pg.10.; lines: 247-248).
4. Lines 315-326. Usually, when p values are more than 0.05, they are reported as p>0.05.
Response 4: Thank you for spotting this. However, APA style dictates reporting the exact p-value within the text of a manuscript (unless the p-value is less than .001) (for example Reporting Statistics in APA Style (ilstu.edu) ).
Reference:
American Psychological Association. (2020). Publication manual of the American Psychological Association (7th ed.). Washington, DC: Author.
5. There are several spelling and grammar errors in the paper, which require correction (line numbers: 14, 18-21, 95, 100, 105, 136, 216, 222, 242, 305, 307, 308, 356, 416, 446, 466, 476, 490, figures A1-A2 “summ”.
Response 5: Thank you for spotting this! This suggestion improved our paper and the readability of the text. We made a grammatical correction along with the proofreading through the whole manuscript. Since incorporating additional text resulted in different number of lines please take this into consideration while reading grammar and spelling corrections (please see following lines within the revisited manuscript: 14, 18-25, 42-43, 45, 96-97, 100, 102, 145, 225, 230, 314,16-317, 426, 457, 476, 496, 503). In addition, figures A1-A2 have been changed (see pg. 17 and 18.)
6. Definitions of some acronyms are required within the text and tables. Particularly in tables, these should present outcomes without to be required checking the text, therefore, the definitions are necessary.
Response 6: Thank you for this very important note. We agree with the reviewers’ suggestion about acronyms and definitions. Accordingly, definitions are incorporated within the tables (please see descriptions below Table 1 and Table 2) (please see Table 1: pg.8.; line: 240-243; Table 2: pg.9.; line: 262).
Reviewer 3 Report
This study examined the association between functional movement (FM) and adiposity in adolescent population. The design, procedures, data collection, data analysis are appropriate, therefore the results and conclusions are reliable. They provide important information on the potential negative impact of adiposity on
movement quality in general adolescents. These information is practical for professionals in youth sport, physical education, and public health. This is the first study on association between FM and adiposity in the general adolescents with large numbers of participants. It is also the first time in the relevant study to employ the multilevel modelling for prediction of the FM through different indicators of adiposity in mid-adolescents. Except for these strengths, there two concerns:
- Lines 189-194: SES should have multiple indicators. It seems too simple to ask only one question.
- 2.1 Participants: Did the missing of participants recruited influence the representative of the original population?
Author Response
General comment to the Editor and Reviewers:
First of all, thank you for considering our manuscript for revisions to your journal International Journal of Environmental Research and Public Health. All suggestions from the Editor and Reviewers are accepted and incorporated into the manuscript. Also, content changes can be seen within the function ‘track changes’ while some grammatical changes are not explicitly highlighted due to the number of errors through text which could impair the flow and readability of the text. Below are specific answers to the Editor and Reviewers comments.
Reviewer #3
Comments and Suggestions for Authors
This study examined the association between functional movement (FM) and adiposity in adolescent population. The design, procedures, data collection, data analysis are appropriate, therefore the results and conclusions are reliable. They provide important information on the potential negative impact of adiposity on movement quality in general adolescents. These information is practical for professionals in youth sport, physical education, and public health. This is the first study on association between FM and adiposity in the general adolescents with large numbers of participants. It is also the first time in the relevant study to employ the multilevel modelling for prediction of the FM through different indicators of adiposity in mid-adolescents. Except for these strengths, there two concerns:
Response to general comment: Thank you for your overall comment about our study and for recognizing the novelty of our research!
1. Lines 189-194: SES should have multiple indicators. It seems too simple to ask only one question.
Response 1: Thank you for spotting this issue within the ‘Material and Methods’ section. The authors agree with the reviewer's comment that is too simple to ask only one question in order to assess SES. Although the authors of the current research are aware that SES should have multiple indicators, this variable was added as a categorical variable within all multilevel regression models. Therefore, the authors believe that this is sufficient in order to discriminate participants between different categories of economic status.
Additionally, school type and the level were considered in all multilevel regression models which is well discussed and described in the 'Results' section. Since the type of school (grammar/vocational/private) could be considered as the indicator of SES, this could further emphasize our adjusted analyses and results given by multilevel modelling within the population of the mid-adolescents.
1. 2.1 Participants: Did the missing of participants recruited influence the representative of the original population?
Response 2: Thank you for this important comment. We conducted additional analyses and results are reported in the revisited manuscript (pg.4.; lines: 122-129). Also, the main findings can be seen in the text below:
To examine if the representativeness of the CRO-PALS sample has been preserved in the subsample selected for FMS assessment, we compared 652 participants from the current study to the rest of the CRO-PALS participants. These analyses indicated comparable values of moderate-to-vigorous physical activity and sum of four skinfolds (p=0.8 and p=0.6, respectively), while values of the sum of four skinfolds, waist circumference, and hip circumference were slightly different between the participants from the current study compared with the rest of participants (BMI: 20. 7 vs 22.3, respectively, p < 0.001; Waist circumference: 70.9 vs 72.8, respectively, p= 0.008; Hip circumference: 95.9 vs 97.8, respectively, p=0.002).